# Reporting of Hybrid Data and the Difficulties with Cross-Discipline Research Techniques

**DOI:** 10.3390/proteomes8040035

**Published:** 2020-11-23

**Authors:** Matthew B. O’Rourke, Matthew P. Padula

**Affiliations:** 1Bowel Cancer & Biomarker Lab, Royal North Shore Hospital, Kolling Institute, Northern Clinical School, Faculty of Medicine and Health, The University of Sydney Level 8, St Leonards, NSW 2065, Australia; matthew.orourke@sydney.edu.au; 2School of Life Science and Proteomics Core Facility, Faculty of Science, The University of Technology Sydney, Ultimo, NSW 2007, Australia

**Keywords:** MALDI-MSI, tissue imaging, hybrid data, complex analysis, expertise

## Abstract

Peer review is the way in which we, as scientists, criticise, check, and confirm the findings of our colleagues. The process of peer review relies on individuals in all fields applying their particular expertise and determining if they agree with the findings submitted for publication. In recent years, there has been a significant rise in the number of manuscripts submitted for publication that draw from a range of disparate and complementary fields. This has created the curious situation where an expert may be requested to review a manuscript that is only partially within their immediate field of expertise. The issue that arises is that, without full knowledge of the data, techniques, methodologies, and principles that are presented, it is difficult for reviewers to make properly informed decisions, especially when it can take an entire career to reach that specific level of expertise in a single field. From this perspective, we explore these issues and also provide a commentary on how peer review could evolve in the context of a changing cross-disciplinarily-focused scientific landscape.

When each scientist begins their career, it is common to find a particular field of interest and then stick to it. Specificity to a particular field is fundamentally how science itself is organised and can be seen in the way in which it is currently taught and the way in which is it also published and represented. In terms of science education, this can be expressed in terms of an “hourglass” model, whereby science is taught from early childhood, covering a wide range of topics [1] and gradually focusing down during undergraduate and post-graduate studies to focus on a very specific subtopic [2]. Once graduated, post-doctoral employment forces the individual to expand their scope of knowledge until, at the eventual professorial level, their understanding of science is very broad and may laterally extend to completely different fields than the subject of their doctoral education (see Figure 1A). The corollary of this is that, as breadth of knowledge expands for the individual, the depth of that knowledge becomes shallower (Figure 1B). This is reflected in the role the scientist performs as they make the change from the bench to the office as their career progresses.

The scope of avenues of current science are incredibly vast, so a separation of each field based on the inherent skills and ideas specific to each field seems logical. However, a difficulty can form when scientists begin to diversify and combine multiple fields into single research projects.

Funding bodies are encouraging this practice in an effort to expand the potential impact of resulting research while also enabling the results to be published in journals that appeal to very wide readerships.

This has created the rise of the multidisciplinary team with diversified researchers who are experts in their particular scientific niche being supervised by someone at a professorial level who is capable of seeing the “big picture” direction of the research [3].

This idea has been explained through the recent emergence of the idea of “systems biology” (ironically now its own field of investigation) that aims to bring together a host of “omic” style investigations to form coherent biological stories [4]. Proteomics, lipidomics, genomics, metabolomics, and transcriptomics can all be combined to create holistic datasets that aim to fully understand biological problems from multiple perspectives. The principle that drives this integration of the sciences is simple: biology is complex, living systems are made up of a multitude of interacting parts, and the interaction needs to be understood in order for meaningful treatments with few side effects to be made available.

The systems biology approach seems to be the future of research in life sciences. However, fledgling research is still being met with the hurdle of who precisely is knowledgeable enough to understand and adequately peer review the resulting data so that it can be published. This is further compounded by a lack of tools or accepted data representation formats that are able to integrate disparate datasets in a way that can be understood by a wide audience.

Traditional publication avenues involve the opinion of experts who critique the validity of the experiments performed and the quality of the execution of the procedures used to generate the data. The data itself is then scrutinised and assessed with the help of statistics and other applicable benchmarks. The assessor then makes a ruling based on their own expertise, deciding whether the article is worthy to be published (an extensive investigation of peer review can be found in Bornmann, L. [5]). In fields that confine themselves to particular methods and understood workflows, this process works almost flawlessly (barring errant disagreement between learned individuals). The issue begins to arise when particular workflows are concerned with several highly disparate fields of expertise. An example that can be immediately brought forward is the field of Mass Spectrometry Imaging (MSI).

Our own laboratory has published very actively in this field and, in our experience, publishing in this field is especially difficult. MSI is a hybrid field in science that combines some aspects of histology with aspects of chemistry and then produces data commonly seen in microscopy using mass spectrometry. After casual inspection, it becomes clear that deciding precisely where to publish this kind of data is a challenge.

MSI uses a Matrix-Assisted Laser Desorption/Ionisation (MALDI) mass spectrometer to create spatial maps of targeted biomolecule classes. The spatial maps take the form of 2D images that show the location and relative intensity of a given molecule [6]. The dynamic range of the instrument allows for the analysis of virtually any class of biomolecule without the need for antibodies, elemental tags, dyes, fluorophores, or radio isotopes. Though this may seem straightforward, the issue becomes the translation of mass spectrometric data—a complex list of numbers—to literal pictures, and how these pictures need to be interpreted.

When MSI data are reviewed, there are usually one of two responses. If the article is sent to an expert in mass spectrometry, the issue of statistics is often raised, since the proper application of statistics is the gold standard for data analysis in mass spectrometry. This however, is countered by the fundamental question of what should be analysed and which statistical tests should be used. On the other side, when a manuscript is reviewed by a histologist, then there is often a series of technical issues raised surrounding the pixelated nature of the data and the need to perform positive controls with antibodies or some other form of microscope-compatible tag or stain. This fundamental disagreement between the sciences creates a curious situation of papers that are MSI-focused but also contain additional data that have only been added to satisfy reviewers and editors.

In 2017, we published a paper concerned with the dangers of cropping MSI images and how this can lead to inaccuracy and the potential misrepresentation of data [7]. Our motivation behind this manuscript was a request from a reviewer of a previous manuscript of ours that suggested that the data we presented were untidy and needed to be edited. While the review process is blind, it does underline an inherent bias that comes from reviewing hybrid data from the perspective of a single field.

When viewed from the perspective of microscopy, MSI data is low-resolution, pixelated, and does not allow the visualisation or distinction of the whole tissue. By all counts, the data is not of a publishable quality. Conversely, from a mass spectrometry perspective the mass resolution is very low and the number of detected/detectible molecules and the confidence that molecular IDs can be properly assigned is also rudimentary at best. Again, this indicates that the data are either unpublishable or require significant additional validation.

The receiving of disparate sets of negative feedback with very little overlap was profound and, speaking from the perspective of an early-career researcher, somewhat disheartening. The choices we were left with were twofold: either argue that reviewers do not possess the skill to understand the data we had generated (an arrogant argument at best), or modify the data accordingly and potentially commit academic misconduct by doctoring primary data. Thankfully, neither option was required and a middle ground was reached between the managing editor and our team. However, this does exemplify the issue that, when non-“traditional” scientific methodologies are utilised, there should also be a change in practice in how these manuscripts are reviewed.

After publishing in this field for the last six years and suffering the pitfalls of constantly explaining the context in which a given manuscript should be reviewed, it becomes apparent to us that a set of objective standards are needed as an aid. Rather than a benchmark of what constitutes good and bad data, such a standard would be more of a guide that would give context for the reviewer.

For example, in Figure 2 we have an image generated in the MSI of a mouse hindbrain (m/z 6190.29) By all accounts, it is of low resolution; individual pixels are visible and there is no interpolation or smoothing applied, making it seem like it has been artificially zoomed. The lack of resolution could easily be explained by simply stating that: *“this image is at a resolution of 1000 × 1000 pixels”* and that *“this is the result of the laser diameter we used when generating the image which cannot be changed”* (for a full description as to how this type of data was generated, see O’Rourke et al. [8]). However, the reality of how to scrutinise this data effectively is far more complex; requiring that certain factors be taken into account while others must be disregarded as inappropriate.

If we apply the current standards for data interpretation from several image-based fields, we begin to see a disparity between what is expected and what the reality is.

When looking from the perspective of the histologist or the microscopist, images such as these can indicate very low-resolution optical sampling or post-acquisition processing/compression, which are considered to be unethical image representations [9]. The issue is that “resolution” in MSI is a term that refers to a measure of the distance between pixels which directly correspond to the size of the laser used to create the data [7]. The term “image” is in fact more of a colloquialism than a literal description, since the images are digital constructs that are derived from the mass spectrometric data that is collected, with each pixel being an X.Y coordinate where data were acquired by the MALDI laser. With that knowledge in mind, it then becomes obvious that the image is not actually poor quality and is merely a digital representation.

However, if we were to view this data from the perspective of a mass spectrometry expert or person well versed in proteomics techniques, there is a shift towards focusing on the integrity of the data and experimental design with considerations such as sample size and the quality of the mass spectrometry data rather than focusing on what the images show. Sets of standards have been created to help guide which information authors should include; however, to date they do not include MSI [10].

To further complicate these interpretation misunderstandings, when viewing what has been provided, the skilled reviewer would also need to consider that which is not included. What is unsaid is often as important as what is said, and only true experts will be able to identify key details that may be omitted. For example, if the above image was acquired on a Bruker UltrafleXtreme MALDI-MS in linear TOF mode with an 85% laser power and detector gain set to 99×, this may seem reasonable to the biologist but to the technologist this indicates that the instrument is actually close to requiring a replacement mass detector and, as such, none of the data can be trusted. Furthermore, if the images that are displayed are in a low m/z range of 800–2000, the question should be asked as to why this was not performed in reflector TOF mode, and where the accompanying spectra that demonstrate that the mass resolution is sufficient to accurately discern the mass of the analyte being observed are.

With all this in mind, there exists a real chance that if a person who is not well versed in the specifics of the technology or technique were to review the above data, then the above considerations could be wholesale missed and the work may be published when it perhaps should not be. Conversely, in our own experience we have often been asked to provide evidence that the signal we see is not noise and is in fact analyte. The irony of this request is that it is based on an assumption that the imaged area is the tissue space (Figure 3A), whereas the imaged area is in fact the total visible area (Figure 3B) and the lack of data outside the tissue is the result of such an intense signal that the noise appears black. If a scale bar were added to the image, then experience would tell us that the scale bar referred to A and not B. Only a person skilled in MSI would be aware that the reverse is true.

With all of the above in mind, the question then becomes “how do we as a community go about improving the quality of our peer review”, especially in an age where technology is becoming more highly specialised, forcing many of us into sub-specialties while at the same time calling upon us to critically evaluate the work of our colleagues in other sub-specialties. This remains a hard question, but it could be argued that, as multidisciplinary studies become more the norm, the process for peer review should be modified to accept that fact that, while one may be an expert in proteomics, they may not be an expert in genomic bioinformatics. This change could be critical, as it has been argued that the current peer review processes limit creativity amongst researchers [11,12], and creativity is what multidisciplinary research encourages.

The first step to addressing the issues we point out is to ensure and document the adequate training of the individuals that review manuscripts. Although not investigated or reported, it is likely that many reviewers have not undertaken any formal training in reviewing manuscripts in the same fashion that there is no formal training for marking student work or theses, just the accumulation of experience. The lack of competent peer reviewers has been raised previously as a concern [12,13]. Many journals are aware of this and already supply detailed guidelines and resources for the peer-review process (https://plos.org/resources/for-reviewers/), but there is no record of whether reviewers have consulted these materials. It is the author’s experience that, when invited to review a manuscript, the reviewer is asked to nominate their expertise in very broad terms. To address our concerns, a reviewer that has not been used previously by a journal or publisher should be subjected to examination to determine their suitability and ability to review, and this could be made available on a register to other journals. Not only would this improve quality, but it would provide a repository of reviewers for editors to access.

A new modern approach to peer review could aim to capture expertise from all fields that are contained within a manuscript and call upon those individuals to review the work as a whole but also to recognise that what they review must be contextual to their own knowledge. Examples of this are already present, whereby some journals now give reviewers the option to request that manuscripts be reviewed by a statistician in addition to their own feedback. This approach could easily be expanded to incorporate specific review by experts in highly technical fields such as mass spectrometry, artificial intelligence development, computational mathematics, programmers, and other technological fields and, if performed correctly, would ensure that scientific review is actually performed by peers. Thus, it is the responsibility of editors to ensure that the reviewers they engage can demonstrate their suitability to review and that the reviewers have clarity about the expectations of their review.

## Figures and Tables

**Figure 1 proteomes-08-00035-f001:**
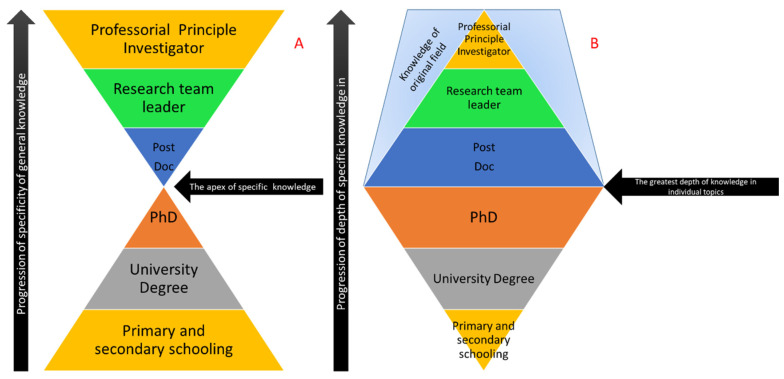
The inverse correlation of specificity of knowledge (**A**) and depth of knowledge (**B**). As the individual progresses towards a more senior position, the breadth of the individual’s knowledge increases, albeit with a shallower understanding. As the original field of knowledge expands, the depth of that information also narrows, however this is only slight.

**Figure 2 proteomes-08-00035-f002:**
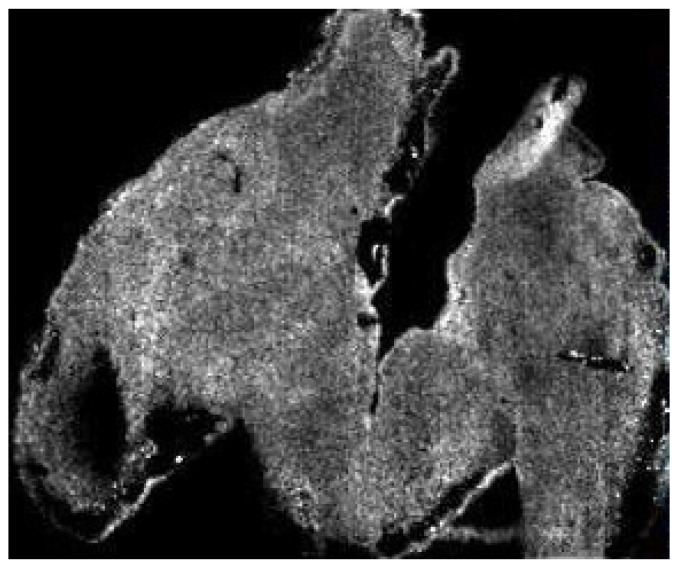
A representative image created with Imaging Mass Spectrometry (MALDI-MSI).

**Figure 3 proteomes-08-00035-f003:**
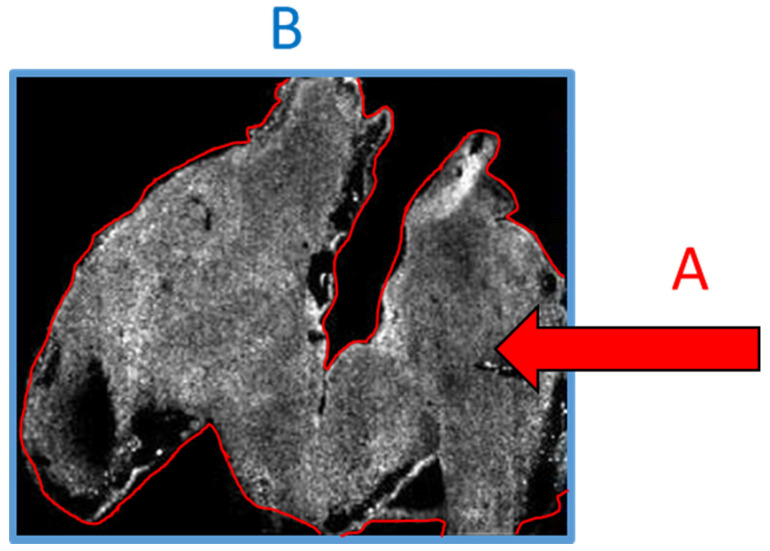
The area actually sampled by the mass spectrometer is the whole image (**B**—blue border) not just the area visible in grey (**A**—red border).

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
