# Peer review of "Reporting of Hybrid Data and the Difficulties with Cross-Discipline Research Techniques"

_proteomes, 2020, doi:10.3390/proteomes8040035_

Round 1

Reviewer 1 Report

This paper makes an important contribution to the reviewing of interdisciplinary manuscripts:
that hybrid data presented in such manuscripts may be approached by very different perspectives
by reviewers from the disparate disciples, applying disparate objectives and criteria.

The specific examples of such data that were persented are quite effective at exposing these concerns. The MSI example
is particularly informative to the argument being made by the paper. Similarly, the paragraph on page 4
discussing the perspective of a mass spectrometry expert made the point very well.

Figure 3 was confusing for this reviewer. There are several white areas in the whole page, and
the red arrow labeled "A" actually points to a gray are of the image, above a white dot. It is the
opinion of this reviewer (who is not at all an expert in any of the relevant fields) that
the final paragraph on page 4 is much clearer than Figure 3, which in fact confuses things.

There was no solution proposed, other than "a new modern approach to peer review shouyld aim
to capture expertise from all fields that are contained within a manuscript". THe last paragraph of the
paper was quite unsatisfying.

Clearly identifying this problem is in the opinion of this reviewer sufficient, though the
paper would be greatly improved if some specific reviewing standards or processes were at least
proposed. The paper, in giving some helpful illustrative examples, could build on those
examples to provide a few approaches.

Finally, the paper is well written, but suffers from haste in preparation, as indicated
by the following list.

Page 1

* "the process" The first letter of a sentence should be capitalized.
* "submitted for publication which" the authors should study the rules for "which" versus "that"
* "Especially" is a sentence fragment
* "evolve in the context of an evolving": it is awkward to repeat a word
* "Once graduated, Post-doctoral" no need to capitalize "Post"

Page 2

* "and aims to": "that aims to" makes the point better
* "transcriptomic" -> "transcriptomics"
* "is simple; biology" a colon would make the point better
* "life sciences, however": sentence runon
* "expertise; deciding": sentence fragment; replace with a colon

Page 3

* "data; a complex": sentence fragment, replace with colon
* "phenomena... profound": how can phenomena itself be profound: it just sits there... Perhaps
what is meant is that the phenomena has profound implications
* "Either..." is a sentence fragment
* "last 6 years": small numbers should be spelled out: "six"

Author Response

Our responses are in italics.

This paper makes an important contribution to the reviewing of interdisciplinary manuscripts: that hybrid data presented in such manuscripts may be approached by very different perspectives by reviewers from the disparate disciples, applying disparate objectives and criteria.

The specific examples of such data that were persented are quite effective at exposing these concerns. The MSI example is particularly informative to the argument being made by the paper. Similarly, the paragraph on page 4 discussing the perspective of a mass spectrometry expert made the point very well.

Figure 3 was confusing for this reviewer. There are several white areas in the whole page, and the red arrow labeled "A" actually points to a gray are of the image, above a white dot. It is the opinion of this reviewer (who is not at all an expert in any of the relevant fields) that the final paragraph on page 4 is much clearer than Figure 3, which in fact confuses things.

We have modified the figure for clarity.

There was no solution proposed, other than "a new modern approach to peer review shouyld aim to capture expertise from all fields that are contained within a manuscript". THe last paragraph of the paper was quite unsatisfying.

Clearly identifying this problem is in the opinion of this reviewer sufficient, though the paper would be greatly improved if some specific reviewing standards or processes were at least proposed. The paper, in giving some helpful illustrative examples, could build on those examples to provide a few approaches.

We welcome the reviewer's invitation to further express our opinions and have revised the final paragraphs to include potential solutions.

Finally, the paper is well written, but suffers from haste in preparation, as indicated by the following list.

All of the following points have been addressed in the revised manuscript. We thank the reviewer for pointing them out and apologise for our lack of rigor.

Page 1

* "the process" The first letter of a sentence should be capitalized.
* "submitted for publication which" the authors should study the rules for "which" versus "that"
* "Especially" is a sentence fragment
* "evolve in the context of an evolving": it is awkward to repeat a word
* "Once graduated, Post-doctoral" no need to capitalize "Post"

Page 2

* "and aims to": "that aims to" makes the point better
* "transcriptomic" -> "transcriptomics"
* "is simple; biology" a colon would make the point better
* "life sciences, however": sentence runon
* "expertise; deciding": sentence fragment; replace with a colon

Page 3

* "data; a complex": sentence fragment, replace with colon
* "phenomena... profound": how can phenomena itself be profound: it just sits there... Perhaps
what is meant is that the phenomena has profound implications
* "Either..." is a sentence fragment
* "last 6 years": small numbers should be spelled out: "six"

Reviewer 2 Report

This is a very valuable comment from inside the field. As authorship is ever-more divided into several roles, each quite specific and distinct, it seems only sensible to also divide peer review into specific roles as the underlying epistemes require. The authors place this issue very well in a general outline and connect it to a specific problem of theirs. This also touches on issues of responsibility that not only remains with the authors but pertains to reviewers and publishers as well. Great to see the debate on peer review gaining traction in specific fields.

Disclaimer: as I’m not trained in proteomics or natural sciences in general, I cannot comment on the field-specific example that makes up most of the article. Ironically, this is just that issue that the authors refer to. However, the way the authors incorporate the example in their argument and the argument itself are straightforward and seem useful as comment to their field.

My only request for change concerns the trajectory of knowledge and careers: the first figure and the corresponding explanation. This is an interesting representation. However, it seems to be a bit shallow in respect to the notion of the depth of knowledge (side B). The combination of side A and B suggests that depth of knowledge unequivocally decreases, which cannot be the case. I consider that you may not be able to generalise this as disciplines behave quite differently in this respect, but a striking commonality may be that, as an individual researcher, at least in one field you retain a depth of knowledge similar to that acquired during a PhD. Of course, the field or the focus may change and, as you outline through side A, the specificity decreases as the researcher gains knowledge about other fields as well. This may be seen as a dialectics of academic careers. But as a PI, it is to hope that you have the knowledge that is at least as deep as that of your PhD students. So the two sides are not particularly an inversion; they are so up to the PhD. After the PhD, side B needs to differentiate, retaining depth in a topic (or developing depth for another during a post doc), but acquiring further topics which never achieve the epistemic level of the earlier focus.

Author Response

This is a very valuable comment from inside the field. As authorship is ever-more divided into several roles, each quite specific and distinct, it seems only sensible to also divide peer review into specific roles as the underlying epistemes require. The authors place this issue very well in a general outline and connect it to a specific problem of theirs. This also touches on issues of responsibility that not only remains with the authors but pertains to reviewers and publishers as well. Great to see the debate on peer review gaining traction in specific fields.

Disclaimer: as I’m not trained in proteomics or natural sciences in general, I cannot comment on the field-specific example that makes up most of the article. Ironically, this is just that issue that the authors refer to. However, the way the authors incorporate the example in their argument and the argument itself are straightforward and seem useful as comment to their field.

My only request for change concerns the trajectory of knowledge and careers: the first figure and the corresponding explanation. This is an interesting representation. However, it seems to be a bit shallow in respect to the notion of the depth of knowledge (side B). The combination of side A and B suggests that depth of knowledge unequivocally decreases, which cannot be the case. I consider that you may not be able to generalise this as disciplines behave quite differently in this respect, but a striking commonality may be that, as an individual researcher, at least in one field you retain a depth of knowledge similar to that acquired during a PhD. Of course, the field or the focus may change and, as you outline through side A, the specificity decreases as the researcher gains knowledge about other fields as well. This may be seen as a dialectics of academic careers. But as a PI, it is to hope that you have the knowledge that is at least as deep as that of your PhD students. So the two sides are not particularly an inversion; they are so up to the PhD. After the PhD, side B needs to differentiate, retaining depth in a topic (or developing depth for another during a post doc), but acquiring further topics which never achieve the epistemic level of the earlier focus.

We have modified the figure as suggested. Thank you for the feedback.